# HARE: Explainable Hate Speech Detection with Step-by-Step Reasoning

**Warning: This paper contains examples of content that is offensive and may be upsetting.**

**Yongjin Yang**[*]   **Joonkee Kim**[*]   **Yujin Kim**[*]   **Namgyu Ho**
**James Thorne**[†]   **Se-young Yun**[†]
KAIST AI
{dyyjkd, joonkeekim, yujin399, itsnamgyu, thorne, yunseyoung}@kaist.ac.kr

## Abstract

With the proliferation of social media, accurate detection of hate speech has become critical to ensure safety online. To combat nuanced forms of hate speech, it is important to identify and thoroughly explain hate speech to help users understand its harmful effects. Recent benchmarks have attempted to tackle this issue by training generative models on free-text annotations of implications in hateful text. However, we find significant reasoning gaps in the existing annotations schemes, which may hinder the supervision of detection models. In this paper, we introduce a hate speech detection framework, **HARE**, which harnesses the reasoning capabilities of large language models (LLMs) to fill these gaps in explanations of hate speech, thus enabling effective supervision of detection models. Experiments on SBIC and Implicit Hate benchmarks show that our method, using model-generated data, consistently outperforms baselines, using existing free-text human annotations. Analysis demonstrates that our method enhances the explanation quality of trained models and improves generalization to unseen datasets. Our code is available at https://github.com/joonkeekim/hare-hate-speech.git.

## 1 Introduction

The increase in the use of online media has intensified the exposure to hate speech, prompting the need for effective detection systems (Schmidt and Wiegand, 2017; Fortuna and Nunes, 2018). While early works have been limited to the classification of explicit hate speech (Caselli et al., 2020; Mathew et al., 2021), recent works have drawn our attention to implicit forms of hate speech which are more prevalent, yet subtle. (Jurgens et al., 2019).

To tackle these nuanced forms of hate speech, it is important for systems to not only identify hate speech but also provide interpretable explanations

---
[*]equal contribution   [†]corresponding authors

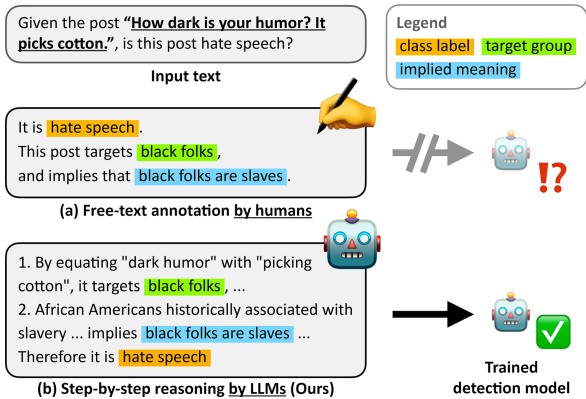

Figure 1: **HARE uses large language models (LLMs) to generate hate speech explanations step-by-step**. (a) Recent benchmarks on understanding hate speech provide free-text annotations on the implications of hate speech, but gaps in reasoning hinder the supervision of generative detection models. (b) We propose the use of LLMs to fill in the gaps and enable detection models to understand and explain hate speech.

(Liu et al., 2019). This can help mitigate distributional biases inherent in simple classification, allowing people to understand and reason about the potential harms of hateful text (Sap et al., 2019b). Explanations can also improve the transparency of content moderation on social media (Gillespie, 2018).

Recent works on hate speech understanding (Sap et al., 2019b; ElSherief et al., 2021; Huang et al., 2022) have considered training autoregressive language models to generate underlying explanations on hate speech. The models are trained on human-written free-text rationales such as implied statements and targeted groups. However, despite the use of novel benchmark datasets, i.e., SBIC (Sap et al., 2019b) and Implicit Hate (ElSherief et al., 2021), the trained models struggle to generate detailed and comprehensive explanations. Moreover, we observe that the provided rationales give marginal improvement to detection performance under joint training.

A potential cause of the limited supervision provided by existing annotations on understanding and explaining hate speech may be the existence of critical gaps in reasoning. For example, as shown in Figure 1, the implied statement of the post "*How dark is my humour? It picks cotton*" is annotated as "*black folks are slaves*", in SBIC. To understand this implication, one must understand that "dark" implies "black folks", and the phrase "picks cotton" relates to the historical background of African Americans. While this may be obvious to human annotators, language models are known to lack societal knowledge and commonsense reasoning skills to understand these nuances (Talmor et al., 2019; Li et al., 2022; Choi et al., 2023). This leaves a significant gap between the training objectives of classification and generating annotated implications, which may harm supervision (Wiegreffe et al., 2021b; Wang et al., 2023a).

Drawing inspiration from the reasoning capabilities of large language models (LLMs) improved with chain-of-thought (CoT) reasoning (Wei et al., 2022), we present our novel approach "Explainable **HA**te Speech Detection with Step-by- Step **RE**asoning (**HARE**)". We leverage LLM-generated free-text rationales using CoT prompts to fill in the gaps of reasoning in existing hate speech annotations and enhance supervision of generative detection models. To create these rationales, we propose two approaches: (1) adopt CoT prompts to create comprehensive rationales that align with the given texts and (2) incorporate existing human annotations from benchmarks in the CoT prompts to bridge the logical gap between the input text and human annotations. When tested on the challenging SBIC and Implicit Hate datasets, our approach outperforms standard fine-tuning with given human annotations and provides enhanced explanations behind the detection results.

## 2 Method

### 2.1 Preliminaries

The task of hate speech detection can be framed as a generative task that inputs the text $P$ and outputs a prediction class $C$, formulated as $p(C|P)$, indicating whether the speech is classified as "hate" or "not hate".[1] Furthermore, by incorporating human-written rationales of the target groups $T$ and implied statements $I$ of hate speech from the SBIC

---

[1]We refer to the "offensive" category in SBIC as "hate" class. In the experiment, we use the wording "offensive" as is.

and Implicit Hate datasets, the task can also be formulated to sequentially output $C$, $T$, and then $I$, as $p([C; T; I]|P)$. Hence, the model trained with human-written annotations is designed to generate $C$ and then provide explanations using annotations $T$ and $I$. However, due to a logical gap between the speech $P$ and the annotations $T$ and $I$, training a model with these annotated rationales does not significantly enhance the model's ability to comprehend hate speech.

Chain-of-thought prompting (Wei et al., 2022) refers to a prompting method that generates a chain of reasoning to derive answers. Kojima et al. (2022) introduce a method of including the phrase "Let's think step by step" at the end of the prompt to generate reasoning in a zero-shot setting.

### 2.2 Fr-HARE & Co-HARE

In this paper, we propose **HARE**, a framework that fine-tunes using rationales generated by LLMs through CoT prompts. By leveraging these rationales, the model is trained to discern whether a text is classified as hate speech or not. This approach of utilizing the LLM-generated rationales not only improve the model performance as in previous works (Ho et al., 2022; Wang et al., 2023a) but also enables the model to generate more plausible explanations. Subsequently, we introduce two variants that employ different methods for extracting rationales from LLMs.

**Fr-HARE** Our first strategy, **Fr**ee-**HARE** (**Fr-HARE**), extracts rationales using LLM from the LLM without referencing the annotated rationales. In line with Kojima et al. (2022), we employ zero-shot prompts to prompt the LLM to extract rationales, as follows:

> Determine whether the following post is offensive, and explain why.
> Post: $P$
> Answer: Let's explain step by step.

From the given prompt above, both the rationale $R$ without any logical leap and the classification result $C$ are generated. Then, we utilize the generated rationale $R$ by training the model to sequentially generate $C$ and $R$, represented as $p([C; R]|P)$. Utilizing these rationales enhances the model's ability to comprehend the context of hate speech, thus leading to explanations that are more intelligible to humans.

**Co-HARE** We also propose a second strategy, **Co**nditioned-**HARE** (**Co-HARE**), which utilizes human-written rationales to formulate CoT prompts. **Fr-HARE** generates complete rationales entirely from scratch, disregarding human annotations included in the datasets. Although **Fr-HARE** offers the advantage of being universally applicable to any hate speech dataset, the LLM-generated rationales might lack consistency due to wide variation in reasoning sequences. To address this inconsistency, **Co-HARE** integrates human-written rationales about target groups $T$ and implied statements $I$ into the CoT prompt as follows:

> Determine whether the following post is offensive, and explain why. Provide a brief, step-by-step explanation of how the post targets the specified group and how it leads to the implied statement provided.
> Post: $P$
> Target: $T$
> Implied statement: $I$
> Answer: Let's explain step by step.

Then, we train the model using extracted rationales, as in **Fr-HARE**. For samples labeled as "not hate" that do not include human-written rationales, we apply the prompt used in **Fr-HARE**. While **Co-HARE** requires human-written rationales, it generates rationales that are more tailored to the specific requirements and features of the dataset, due to its guided nature. Therefore, the model trained with **Co-HARE** can provide explanations that align more closely with the forms of rationales that humans construct.

**Details of HARE** Once we have extracted the rationales from the LLMs, we follow the approach of Kojima et al. (2022) to have the LLMs predict the class. Specifically, we employ a two-stage extraction process. In the first stage, we extract both the class $C$ and the rationale $R$ from the LLMs using our **HARE** method, represented as $p([C; R]|P)$, as previously outlined. In the second stage, we prompt the LLMs again, this time to predict the class $C$ given the extracted rationales $R$ and the post $P$, denoted as $p(C|R, P)$. During fine-tuning on hate speech datasets, if the predicted class $C$ coincides with the true answer $C$, we concatenate $C$ with the extracted rationale $R$. If the predicted labels are incorrect, the models are solely trained to predict the class $C$. Furthermore, following the findings of Ho et al. (2022), we generate multiple distinct rationales to facilitate the learning process.

# 3 Experiments

## 3.1 Experimental Setup

We utilize SBIC and Implicit Hate datasets for our fine-tuning experiments. Our models are trained to classify the offensiveness and hatefulness of posts, using SBIC and Implicit Hate, respectively. It is noteworthy that in our Implicit Hate experiments, we combine both the explicit and implicit hate classes into a single "hate" category. We set up baselines with two families of models: $C$, a model trained exclusively for classification, and $C+T+I$, a model trained using human-written rationales. For **Fr-HARE** and **Co-HARE**, by using gpt-3.5-turbo-0613 that is known for its reasoning capabilities (Ouyang et al., 2022), we extract four and eight different rationales per each sample in SBIC and Implicit Hate, respectively, following the hyperparameter setting of Ho et al. (2022). Subsequently, we fine-tune the model, setting LLM-generated rationales $R$ and class $C$ as target sequence. For performance evaluation, we measure detection accuracy and compute the F1 score of classification, regarding "hate" as the positive class. We make use of Flan-T5 (Wei et al., 2021) with different model configurations: small, base and large. We also conduct experiments using the large models of T5 (Raffel et al., 2020) and GPT-2 (Radford et al., 2019). A more detailed explanation of our experimental setup can be found in Appendix B.

## 3.2 Results and Discussions

**Do LLM-generated rationales improve detection performance?** Table 1 presents the performance of hate speech detection according to different methods on the SBIC and Implicit Hate datasets. Our strategies **Fr-HARE** and **Co-HARE** consistently exhibit superior performance over other baseline methods, regardless of the model size. This suggests that even though the baseline method is trained using human-written rationales, the more detailed and logically-sequenced LLM-generated rationales of **HARE** can further aid the model in understanding the input text and accurately classifying it as hate speech. Therefore, the results demonstrate that the quality of rationales has a strong impact on classification. Furthermore, the performance of our method consistently improves as the model size increases, in contrast to baselines. This suggests that diverse reasoning becomes increasingly beneficial as scale grows. This notable im-

Table 1: The performance of fine-tuning on SBIC and Implicit Hate dataset with various models and size.

| Model | Method | SBIC | | Implicit Hate | |
|---|---|---|---|---|---|
| | | Acc | F1 | Acc | F1 |
| GPT-3.5- | ZS | 80.06 | 81.75 | 73.58 | 65.66 |
| turbo-0613 | ZS-CoT | 73.48 | 79.07 | 73.98 | 67.19 |
| Flan-T5 small | $C$ | 82.56 | 84.05 | 77.58 | 71.98 |
| | $C+T+I$ | 82.99 | 84.05 | 77.63 | 72.39 |
| | Fr-HARE | 84.18 | 85.18 | **79.33** | 73.29 |
| | Co-HARE | **84.44** | **85.35** | 78.54 | **73.49** |
| Flan-T5 base | $C$ | 82.35 | 83.71 | 78.03 | 72.17 |
| | $C+T+I$ | 82.54 | 84.41 | 79.77 | 73.15 |
| | Fr-HARE | 84.20 | 85.46 | 79.84 | 74.84 |
| | Co-HARE | **84.65** | **85.76** | **80.38** | **75.69** |
| Flan-T5 large | $C$ | 81.70 | 82.84 | 78.42 | 72.92 |
| | $C+T+I$ | 83.48 | 83.70 | 80.14 | 73.10 |
| | Fr-HARE | **85.21** | **86.16** | 80.49 | 74.62 |
| | Co-HARE | 84.93 | 85.57 | **81.49** | **76.71** |
| T5 large | $C$ | 83.03 | 83.53 | 78.79 | 72.50 |
| | $C+T+I$ | 84.23 | 85.21 | 79.61 | 73.80 |
| | Fr-HARE | 85.27 | **86.32** | **81.61** | 75.59 |
| | Co-HARE | **85.35** | 85.93 | 80.98 | **75.88** |
| GPT-2 large | $C$ | 81.39 | 82.68 | 73.32 | 66.68 |
| | $C+T+I$ | 82.80 | 83.43 | 75.95 | 65.25 |
| | Fr-HARE | 83.92 | 85.48 | 78.47 | 71.35 |
| | Co-HARE | **84.64** | **85.67** | **80.07** | **71.58** |

provement with **HARE** is achieved by using only 40$ for each method in our approach, demonstrating that the ability to reason can be effectively trained with rationales from LLMs.

Additionally, while **Fr-HARE** and **Co-HARE** exhibit similar performance, **Co-HARE** has a slight edge in most cases. This is because **Co-HARE** is guided by human-written annotations, which results in better alignment with the setting of the datasets, as we mentioned in Section 2.2. It is also noteworthy that all the fine-tuned models surpass both Zero-Shot (ZS) and Zero-Shot CoT (ZS-CoT, Kojima et al. (2022)) classification performance of GPT-3.5-turbo, indicating that merely employing LLM with CoT prompts is not sufficient to tackle this task.

**Are HARE models more generalizable?** To assess the ability of our methods to generalize across different datasets, we evaluate the models fine-tuned on the SBIC datasets using each method on two distinct datasets, HateXplain (Mathew et al., 2021) and DynaHate (Vidgen et al., 2020). Both datasets encompass forms of explicit and implicit hate. On both datasets, our methods **Fr-HARE** and **Co-HARE** both outperform baseline methods, indicating that our methods enhance the generalizability of the models by improving their reasoning

Table 2: Cross Evaluation results on HateXplain (Mathew et al., 2021) and DynaHate (Vidgen et al., 2020). We utilize Flan-T5-large fine-tuned on SBIC using each method.

| Method | HateXplain | | DynaHate | |
|---|---|---|---|---|
| | Acc | F1 | Acc | F1 |
| $C$ | 64.40 | 74.18 | 64.35 | 67.41 |
| $C+T+I$ | 68.84 | 74.52 | 64.72 | 67.41 |
| **Fr-HARE** | 70.69 | **78.91** | 68.06 | **75.15** |
| **Co-HARE** | **71.62** | 78.52 | **69.98** | 75.01 |

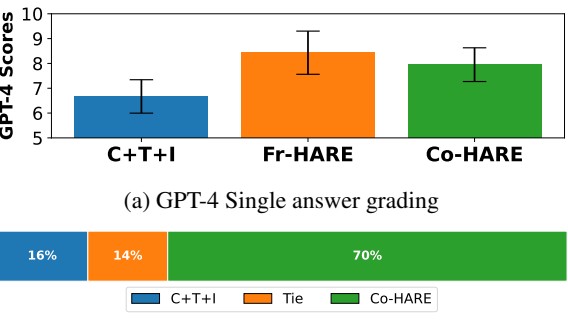

(a) GPT-4 Single answer grading

(b) GPT-4 Pairwise comparison

Figure 2: The result of GPT-4 evaluation following Zheng et al. (2023). The bar and line represent the average scores that range from 1 to 10 and 95% confidence interval, respectively. We utilize Flan-T5-large fine-tuned on SBIC using each method.

ability. Moreover, the comparable performance of **Fr-HARE** and **Co-HARE** suggests that both techniques of bridging logical gaps are effective in tackling hate speech detection.

**Does HARE improve the quality of generated explanations?** To evaluate the quality of the generated explanations, we conduct automated evaluation using GPT-4 (OpenAI, 2023) which has demonstrated its high alignment with human evaluation (Liu et al., 2023; Zheng et al., 2023). We randomly sample 50 "hate" instances from SBIC test set where all methods are correct and apply single answer grading (Zheng et al., 2023) on explanations generated by models trained with $C+T+I$, **Fr-HARE**, and **Co-HARE**. Detailed explanation about GPT-4 evaluation can be found on Appendix C.

The results depicted in Figure 2 (a) show that both **Fr-HARE** and **Co-HARE** are capable of generating higher quality explanations, with **Fr-HARE** exhibiting slightly superior performance, suggesting that the flexibility of **Fr-HARE** leads to a more qualified explanation.

| Post | I like my women like I like my cigars 7 years old and coming from cuba in a burlap sack, |
|---|---|
| Label | (A) Offensive |
| Annotation | girls are not worthy of equal life / girls are property / girls are used for sex / trivializes harm to victims |
| C+T+I | The answer is : (B) Not offensive |
| Fr-HARE | The answer is : (A) Offensive 1.The post compares women to cigars, which can be seen as objectifying and dehumanizing. 2. It also makes a reference to a specific age group, which can be seen as inappropriate and offensive. 3. (…) |
| Co-HARE | The answer is : (A) Offensive 1.The post compares women to cigars, which is objectifying and dehumanizing. (…) 2. The post specifically mentions that the cigars should be "7 years old," which is highly inappropriate and offensive as it promotes child exploitation and sexual abuse. (…) 3. The post also mentions that the cigars should be "coming from Cuba in a burlap sack," which is a clear reference to the sex trafficking scandal that occurred in the United States in the past. (…) |
| Post | "I think a lot, but I don't say much." ~ Anne Frank But lately she's been proven to have not said anything for almost a century |
| Label | (A) Offensive |
| Annotation | jewish people were rounded up and killed in the holocaust / jewish people were victims of a genocide. |
| C+T+I | The answer is : (B) Not offensive |
| Fr-HARE | The answer is : (A) Offensive 1.The post starts with a quote from Anne Frank, a Jewish girl who died during the Holocaust. 2. The quote itself is a statement about her lack of speaking much. 3. (…) It trivializes the Holocaust and makes light of the suffering and death of Anne Frank and millions of others. (…) |
| Co-HARE | The answer is : (A) Offensive 1.The post starts with a quote from Anne Frank, a Jewish girl who wrote a diary during the Holocaust. 2. (…) 3. (…) This post targets Holocaust victims and Jewish folks by making a joke about Anne Frank's lack of speaking for almost a century. It implies that she has been silent or silent about her experiences during the Holocaust. This trivializes the harm and suffering that Holocaust victims endured. (…) |

Figure 3: Model outputs using baseline methods and our framework **HARE**. The samples are from SBIC test set. Note that the answers are abbreviated by (...), and the full context is reported in Appendix D.

As **Co-HARE** is designed to create rationales that align better with human-written rationales, we also conduct pairwise comparison between **Co-HARE** and $C+T+I$ to determine which method aligns better with human-written labels. Figure 2 (b) clearly shows that the rationales generated by the model trained using **Co-HARE** are aligned more to actual human-written rationales than the ones generated by the model trained directly with human-written rationales. This also highlights the value of **Co-HARE**, as it aligns better with natural intuition, indicating that both **Fr-HARE** and **Co-HARE** can be utilized for different purposes.

**How does HARE detect and contextualize hate speech?** Figure 3 presents model outputs using **HARE** and baseline method. A more detailed qualitative study of randomly selected samples is available in Appendix D. In the first sample, the post makes light of harming a young girl using phrases like "like I like cigar" and "burlap sack". While our model, **Fr-HARE**, does not explicitly connect the "burlap sack" to the idea of a kidnapped girl, it does recognize the harmful implications towards the girl. Furthermore, **Co-HARE** understands the historical context behind the term "burlap sack". The presence of the positive term "like" and the ab-

| Post | Why do so many white people own pets? Because they can't own people anymore. |
|---|---|
| Label | (A) Offensive |
| Annotation | black people are inferior / black people should be slaves |
| LLM rationale | 1.The post makes a generalization about white people and their ownership of pets. 2. It suggests that white people historically owned people, implying slavery. 3. It implies that white people now own pets as a replacement for owning people. This post can be considered offensive because it perpetuates a harmful stereotype about white people and makes light of a dark and painful history of slavery. It reduces the complex issue of slavery to a simplistic and offensive joke. While humor can be subjective, this post crosses the line by making inappropriate and insensitive comparisons. |

Figure 4: A sample of LLM rationale generated by GPT-3.5-turbo using **Fr-HARE** from SBIC train set.

sence of overtly derogatory words might lead some models to classify the statement as non-offensive. However, our approach appropriately identifies the underlying hateful context.

In the second sample, the post contains hate speech targeting Jewish victims of the Holocaust by referencing Anne Frank. Our model accurately recognizes the historical background of Anne Frank as a Holocaust victim. While the baseline seems to overlook the historical significance associated with Anne Frank, our method correctly identifies her and assumes that the reference constitutes harassment against a Jewish victim, even though there is a slight misunderstanding about the context of "lack of speaking" in **Fr-HARE**.

**Is GPT-3.5 a qualified teacher?** Since our framework is based on distillation of generated rationales from GPT-3.5 to smaller models, it is crucial to verify whether the teacher is qualified. Figure 4 displays rationales produced by GPT-3.5-turbo, which is employed to train the student model. This example illustrates that the LLM not only discerns the hateful nuances towards both white and black individuals, but also offers more detailed explanations compared to rationales written by humans. Notably, it accurately correlates the historical context, associating the word "slaves" with "pets". More analysis of rationales from GPT-3.5-turbo can be found in Appendix D.2.

## 4 Conclusion

In this paper, we present **HARE** framework to improve the ability of the language model to understand hate speech and provide clearer explanations for its decisions. We propose utilizing CoT reasonings extracted from LLMs in two variants to overcome the logical gaps in human-annotated rationales. When fine-tuned on the SBIC and Implicit Hate datasets, our methods achieve superior detection performance and better qualified explanations.

## Limitations

While we assess the quality of explanations generated by HARE using GPT-4, we do not conduct human evaluations, which are crucial for tasks requiring human-readable explanations. The primary reason for this omission is that the hate speech content and its respective explanations could be excessively offensive for annotators and GPT-4 already aligns with the level of inter-human agreement. In addition, the "verbosity bias", characterized by a preference for the longer text of GPT-4 as indicated by (Liu et al., 2023), may also serve as a limitation in our evaluation process.

## Ethics Statement

Predicting whether an online post contains hate-speech is both technically and socially challenging. While methods for automating hatespeech detection have utility in an online platform, it is critical that these are tuned and used appropriately. False-positive errors have potential to censor online speech, further marginalizing specific user groups, for example: use of $n*****$ in AAVE English may be flagged. It is critical to understand specific reasoning behind a classification including deeply social reasons. While language models act as a mechanism to generate reasonable explanations, it is critical that they are used appropriately to prevent them from inadvertently educating users on how to craft more subtle and toxic language. We used automated evaluation metrics in this paper to prevent exposure of toxic language to human annotators. However, real-world usage would require validation that deeply rooted social issues are expressed correctly by these models.

It is also important to note that there might be concerns about the inherent bias in the GPT-3.5 model. While not flawless, GPT-3.5 has demonstrated its impartiality regarding gender, race, ethnicity, and religion by achieving the highest grade on the Harmfulness metric within the FLASK evaluation framework (Ye et al., 2023). Crucially, we only select rationales that align with the ground truth label for training, thereby mitigating biases not in sync with human annotators. Analysis of GPT-3.5-turbo can be found in Section 3 and Appendix D.2.

## Acknowledgement

This work was supported by Institute of Information & communications Technology Planning &

Evaluation (IITP) grant funded by Korea government (MSIT) [No. 2021-0-00907, Development of Adaptive and Lightweight Edge-Collaborative Analysis Technology for Enabling Proactively Immediate Response and Rapid Learning, 90%] and [No. 2019-0-00075, Artificial Intelligence Graduate School Program (KAIST), 10%].

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

## A Related Work

**Hate Speech Detection** Hate speech (Waseem et al., 2017) is a form of language designed to offend a particular individual or groups. In this study, we expand this definition by incorporating the broader concept of offensive language as in (Burnap and Williams, 2016; Ribeiro et al., 2018). Numerous recent works on hate speech detection have delved into providing underlying explanations of prediction on hate speech (Sap et al., 2019a,b; Mathew et al., 2021; ElSherief et al., 2021; Lin, 2022). One line on research focuses on keyword-based explanations (Sap et al., 2019a; Davidson et al., 2019; Mathew et al., 2021; Kim et al., 2022), but this approach often fails to capture implicit hatefulness that is not explicitly present in the text. Another approach involves explanations utilizing external knowledge sources (Sridhar and Yang, 2022; Lin, 2022), but these methods aim to solely improve classification performance. Yet another studies involve training generative models with human-written free-text rationales (Sap et al., 2019b; ElSherief et al., 2021; Huang et al., 2022) present in multiple benchmarks (Sap et al., 2019b; ElSherief et al., 2021). Nevertheless, due to the existence of logical gaps in these human-annotated rationales (Aggarwal et al., 2021; Sun et al., 2022), relying solely on these rationales results in sub-optimal detection and explanation quality. Our proposed **HARE** shows its effectiveness by incorporating LLM-generated rationales, which include logical completeness and abundant explanatory power extracted with our CoT prompting.

**Self-Rationalization** Self-rationalization, a technique where models provide explanations for their predictions, has been extensively studied to make models more understandable and transparent (Marasović et al., 2021; Wiegreffe et al., 2021a,b). Recent studies leverage rationale-augmented exemplars to few-shot prompt LLMs (Wei et al., 2022; Wang et al., 2022; Lampinen et al., 2022), while others fine-tune smaller models using the rationales generated by LLMs (Ho et al., 2022; Wang et al., 2023a; Chan et al., 2023; Wang et al., 2023b). Several works apply their self-rationalization methods to detect hate speech and generate underlying explanations (Marasović et al., 2021; Ludan et al., 2023; Chen et al., 2023). However, they directly use human-written rationales in their framework, which may lack logical completeness. Instead, our method adopts the approach of utilizing the rationales generated by LLMs.

## B Implementation Details

**Training Details** All the experiments utilize an Adafactor (Shazeer and Stern, 2018) optimizer with batch size of 32 and learning rate $\in$ {5e-3, 5e-4, 5e-5} without learning rate scheduling. We train all models for 10 epochs and utilize the checkpoint with the highest validation F1 score to evaluate on the test set. Top-k sampling with $k$=20 is adopted as a decoding method. For dataset split, we use official dataset split for SBIC and random split with 6:2:2 ratio for train, validation, and test sets for Implicit Hate. All our experiments are conducted using a single NVIDIA A6000 GPU.

## C Details on GPT-4 Evaluation

To conduct GPT-4 evaluation, we utilize the prompts proposed by Zheng et al. (2023). For single answer grading, the prompt format for GPT-4 is written as follows:

> [Instruction]
> Please act as an impartial judge and evaluate the quality of the response provided by an AI assistant to the user question displayed below. Your evaluation should consider factors such as the helpfulness, relevance, accuracy, depth, creativity, and level of detail of the response. Begin your evaluation by providing a short explanation. Be as objective as possible. After providing your explanation, please rate the response on a scale of 1 to 10 by strictly following this format: "[[rating]]"
> [Question]
> Determine whether the following post is offensive or not. You must choose one of the options.
> Post: $P$
> Options:
> (A) Offensive
> (B) Not offensive
> [The Start of Assistant's Answer]
> $C, R$
> [The End of Assistant's Answer]

To compare the alignment of generated rationales with human-written rationales, we conduct a pairwise comparison of explanations against ground-truth rationales. These ground-truth rationales encompass the annotated target groups and implied statements from the instruction, and we compare the results. Additionally, to minimize bias from the order of candidate answers, we adopt the approach of Zheng et al. (2023), considering both original and swapped orders of predicted explanations. If $C + T + I$ and **Co-HARE** are chosen alternately, it is deemed a tie. Should one method

be selected following a tie, that method is considered as the chosen one. The prompt format for the pairwise comparison is provided below:

> [Instruction]
> Please act as an impartial judge and evaluate the quality of the responses provided by two AI assistants to the user question displayed below. You should choose the assistant that follows the user's instructions and answers the user's question more accurate. When choosing the assistant, please consider the true answers below:
> Target: $T$
> Implied Statement: $I$
> Your evaluation should consider which response is more similar to the true answers. Begin your evaluation by comparing the two responses and provide a short explanation. Avoid any positional biases and ensure that the order in which the responses were presented does not influence your decision. Do not allow the length of the responses to influence your evaluation. Do not favor certain names of the assistants. Be as objective as possible. After providing your explanation, output your final verdict by strictly following this format: "[[A]]" if assistant A is more accurate, "[[B]]" if assistant B is more accurate, and "[[C]]" for a tie.
> [Question]
> Determine whether the following post is offensive or not. You must choose one of the options.
> Post: $P$
> Options:
> (A) Offensive
> (B) Not offensive
> [The Start of Assistant A's Answer]
> **Answer from one method**
> [The End of Assistant A's Answer]
> [The Start of Assistant B's Answer]
> **Answer from another method**
> [The End of Assistant B's Answer]

## D Qualitative Study

### D.1 Qualitative Study of HARE

Figures 5, 6, 7, and 8 showcase results generated by the fine-tuned Flan-T5-large model using **HARE** and $C+T+I$, based on test samples from SBIC. Although a brief explanation is provided in Section 3.2, we delve deeper with an extended analysis of the 20 examples from our qualitative study. These 20 samples were randomly chosen in proportion to their correct and incorrect predictions across the different methods.

When comparing human-written annotations with **HARE**, it becomes evident that the annotated rationales in SBIC often take the form of implied statements, following a simple Hearst-like pattern (Sap et al., 2019b). Learning from such rationales, which are closely tied to the conclusion, creates a logical gap for the model and makes interpretation challenging for humans. For instance, understanding hate speech without background knowledge references, such as 'burlap sack', can make it difficult to see the connection between the statement "girls are not worthy of equal life" and the provided sentence. Figures 5 and 6 showcase successful cases where models have attempted to bridge this reasoning gap through **HARE**, offering more detailed rationales that encompass the context. Furthermore, these models exhibit capabilities not seen in previous research, such as detecting terms with historical significance (e.g., 'burlap sack' or 'Anne Frank') or common words that may carry hateful connotations (e.g., 'reds'), thus enhancing the intermediate reasoning process.

However, when examining the failure cases in Figures 7 and 8, the results show that **HARE** sometimes fails due to increased sensitivity to potentially harmful terms, thereby classifying them as offensive. While this increased sensitivity can be viewed as a drawback, there are instances, such as with the Alzheimer example, where an expression might be interpreted as hateful depending on the individual. This suggests that **HARE** aims to classify a post as hateful if it could be considered offensive to certain groups. Moreover, considering the David Bread Katz example, it is also challenging for **HARE** to decide if the post is offensive if it post with background that it hasn't encountered, possibly due to a lack of background knowledge regarding the implied shooting incident, illustrating the limitation of LLM distillation.

### D.2 Qualitative Study on GPT-3.5 rationales

When comparing annotations with rationales generated by GPT-3.5, we observe that human-written rationales from SBIC use implied statements that follow simple Hearst-like patterns (Sap et al., 2019b). In contrast, models like LLM (e.g., GPT-3.5) tend to provide detailed, step-by-step explanations, often complemented by relevant social background information, which is immensely beneficial. For example, while earlier rationales might omit mentioning Bill Cosby's conviction of sexual assault, GPT-3.5 explicitly informs us of this fact, greatly enhancing comprehension. A particularly striking example is the "pet" case. While a human annotator perceived it as hate speech targeted at black individuals, GPT-3.5 points out that it could also be used derogatorily against white individuals, thereby emphasizing the potential biases in hate speech detection.

When rationales are categorized and structured, as seen in SBIC, instead of being tailored to individual posts, they may not be sufficient for learning implications. This could explain why, as suggested by Table 2, there's a decrease in generalization for $C+T+I$. Our approach offers aligned rationales for each post at minimal cost, enabling the learning of diverse reasons for potential hate, which in turn leads to enhanced generalization.

| | |
|---|---|
| **Post** | I like my women like I like my cigars 7 years old and coming from cuba in a burlap sack, |
| **Label** | (A) Offensive |
| **Annotation** | girls are not worthy of equal life / girls are property / girls are used for sex / trivializes harm to victims |
| **C+T+I** | The answer is : (B) Not offensive |
| **Fr-HARE** | The answer is : (A) Offensive 1.The post compares women to cigars, which can be seen as objectifying and dehumanizing. 2. It also makes a reference to a specific age group, which can be seen as inappropriate and offensive. 3. The post also mentions Cuban cigars, which can be seen as promoting or glorifying illegal activities. Based on these steps, the post can be considered offensive. It objectifies and dehumanizes women, promotes illegal activities, and includes references to illegal activities. |
| **Co-HARE** | The answer is : (A) Offensive 1.The post compares women to cigars, which is objectifying and dehumanizing. This targets women by reducing them to objects and implying that they are only valuable for their physical appearance. 2. The post specifically mentions that the cigars should be "7 years old," which is highly inappropriate and offensive as it promotes child exploitation and sexual abuse. This targets women by implying that they are sexual objects and should be subjected to the same level of abuse as cigars. 3. The post also mentions that the cigars should be "coming from Cuba in a burlap sack," which is a clear reference to the sex trafficking scandal that occurred in the United States in the past. This targets women by implying that they are only valuable for their sexual appeal and that their worth is solely based on their ability to fulfill sexual desires. Overall, the post is offensive as it objectifies and devalues women, promotes child exploitation, and makes light of a serious and sensitive topic. |
| **Post** | "I think a lot, but I don't say much." ~ Anne Frank But lately she's been proven to have not said anything for almost a century |
| **Label** | (A) Offensive |
| **Annotation** | jewish people were rounded up and killed in the holocaust / jewish people were victims of a genocide. |
| **C+T+I** | The answer is : (B) Not offensive |
| **Fr-HARE** | The answer is : (A) Offensive 1.The post starts with a quote from Anne Frank, a Jewish girl who died during the Holocaust. 2. The quote itself is a statement about her lack of speaking much. 3. The post then makes a comparison by stating that she has been proven to have not spoken anything for almost a century. Based on this analysis, the post can be considered offensive. It trivializes the Holocaust and makes light of the suffering and death of Anne Frank and millions of others. The comparison between her lack of speaking and her lack of speaking is insensitive and disrespectful. |
| **Co-HARE** | The answer is : (A) Offensive 1.The post starts with a quote from Anne Frank, a Jewish girl who wrote a diary during the Holocaust. 2. The quote states, "I think a lot, but I don't say much." 3. The post then adds, "But lately she's been proven to have not said anything for almost a century." This post targets Holocaust victims and Jewish folks by making a joke about Anne Frank's lack of speaking for almost a century. It implies that she has been silent or silent about her experiences during the Holocaust. This trivializes the harm and suffering that Holocaust victims endured. Therefore, the post can be considered offensive as it makes light of a tragic event and trivializes the harm suffered by Holocaust victims. |
| **Post** | Actually I never had too many women until I became a white nationalist . |
| **Label** | (A) Offensive |
| **Annotation** | - |
| **C+T+I** | The answer is : (B) Not offensive |
| **Fr-HARE** | The answer is : (A) Offensive 1.The post states that the person has never had many women until they became a white nationalist. 2. White nationalism is a belief system that promotes the belief that white people are superior to other races or ethnicities. 3. The post implies that the person's lack of women is due to their belief in white superiority, which is offensive and promotes discrimination. 4. The post also implies that the person's lack of women is due to their belief in white superiority, which is offensive and promotes racial superiority. Based on these steps, the post can be considered offensive. It promotes discrimination and discrimination based on race. |
| **Co-HARE** | The answer is : (A) Offensive 1.The post states that the person has never had too many women until they became a white nationalist. This implies that the person believes that white nationalists are superior or more desirable than other racial or ethnic groups. 2. The post specifically mentions that the person has never had too many women before becoming a white nationalist. This implies that the person believes that white nationalists are superior or more desirable than other racial or ethnic groups. 3. The post targets "black folks, asian folks, latino/latina folks, native american/first nation folks" by implying that they are inferior or less desirable than other racial or ethnic groups. This implies that these minority groups are not as good as whites and that they should be excluded or excluded from society. 4. The implied meaning of the post is that minorities are inferior, should be excluded, and should be confined to certain racial or ethnic groups. This is offensive because it promotes discrimination, inequality, and the devaluation of a specific racial or ethnic group. |
| **Post** | i can't stand these ""grown"" ass females 🤦🏽\u200d\u2640\ufe0f🤦🏽\u200d\u2640\ufe0f how you "" grown "" but childish as hell ?! 😩😂🤦🏽\u200d\u2640\ufe0f🤦🏽\u200d\u2640\ufe0f |
| **Label** | (A) Offensive |
| **Annotation** | calls women females / women are childish |
| **C+T+I** | The answer is : (B) Not offensive |
| **Fr-HARE** | The answer is : (A) Offensive 1.The post uses derogatory language by referring to females as "grown" and "childish." This language is offensive as it dehumanizes and insults women. 2. The post also uses emojis that can be interpreted as mocking or belittling the mentioned females. This can be seen as offensive as it belittles and mocks them. 3. The post also uses emojis that can be interpreted as mocking or belittling the mentioned females. This can be seen as offensive as it belittles and mocks them. Based on these steps, the post can be considered offensive. It uses derogatory language, emojis, and mocks a specific group of people. |
| **Co-HARE** | The answer is : (A) Offensive 1.The post uses the term "grown" to describe women, which can be seen as derogatory and offensive. 2. The post uses the term "ass" to describe women, which can be seen as demeaning and disrespectful. 3. The post uses the term "childish as hell" to describe women, which can be seen as demeaning and offensive. Based on these steps, the post can be considered offensive. It uses derogatory language, degrades women, and includes derogatory terms. |
| **Post** | Dunnes strikers to attend funeral - BelfastTelegraph.co.uk Lets hope these reds are n't mistaken for the usual white targets of these jungle savages ! |
| **Label** | (A) Offensive |
| **Annotation** | black folks are savages / savage / violent |
| **C+T+I** | The answer is : (B) Not offensive |
| **Fr-HARE** | The answer is : (A) Offensive 1.The post mentions Dunnes strikers attending a funeral. 2. It then makes a statement about the reds being mistaken for the usual white targets of these "jungle savages." Based on this explanation, the post can be considered offensive. The statement about the reds being mistaken for the usual white targets of these "jungle savages" implies a negative stereotype about a specific racial group, suggesting that they are inherently dangerous or primitive. This kind of language promotes racism and discrimination. |
| **Co-HARE** | The answer is : (A) Offensive 1.The post mentions Dunnes strikers attending a funeral. 2. It then states, "Let's hope these reds aren't mistaken for the usual white targets of these jungle savages!" 3. The term "reds" is a derogatory term for black people, implying that they are inferior or less than human. 4. The post also uses the term "jungle savages" to refer to black people, further dehumanizing them. 5. The post implies that black people are savages, which is a derogatory and offensive stereotype. Overall, the post targets black people by using derogatory language and perpetuating harmful stereotypes. It implies that black people are inferior, savages, and savages. Therefore, the post is offensive. |

Figure 5: Success cases of **HARE**

| | |
|---|---|
| **Post** | Four-year-old boy:: Why did she get to walk in first? Dad:: Because she's a girl. And girls go first. Four-year-old boy:: Why? That's DUMB!' Dad:: Because girls need protecting from strong boys like you. |
| **Label** | (A) Offensive |
| **Annotation** | women are submissive / women are weak / women can't take care of themselves / women need protection |
| **C+T+I** | The answer is : (A) Offensive. The post targets women and implies are weak. |
| **Fr-HARE** | The answer is : (A) Offensive 1.The post starts with a four-year-old boy asking why a girl got to walk in first. 2. The dad responds by saying that girls go first. 3. The four-year-old boy questions why this is dumb. 4. The dad responds by saying that girls need protection from strong boys like the boy. Based on this explanation, the post can be considered offensive. The dad's response implies that the girl's gender determines her ability to protect herself from boys, which is a harmful stereotype. It also perpetuates gender bias and reinforces gender stereotypes. |
| **Co-HARE** | The answer is : (A) Offensive 1.The post starts with a four-year-old boy asking why a girl got to walk in first. The father responds by saying that girls go first. This implies that the boy is being criticized for being able to walk in first because he is a girl. 2. The boy then questions why he got to walk in first, implying that he believes it is foolish. 3. The father responds by saying that girls need protection from strong boys like him. This implies that the boy is being criticized for being weak or weak. Overall, the post targets women by suggesting that they are weak and need protection from men. It implies that women are weak and need to be protected from men. This can be seen as offensive as it perpetuates gender stereotypes and undermines the abilities and strength of women. |
| **Post** | Frederick neichzche embraced the notion that for a race to evolve there had to be a struggle to kill of the weaker elements to keep them from contaminating the gene pool. in the US one could say that there is a breeding program to breed dumb ghetto blacks . |
| **Label** | (A) Offensive |
| **Annotation** | black folks are inferior / black folks are trashy. / black folks are unintelligent. / black folks aren't smart / black people are dumb / blacks breed too much |
| **C+T+I** | The answer is : (A) Offensive. The post targets black folks and implies black folks are ghetto. |
| **Fr-HARE** | The answer is : (A) Offensive 1.The post mentions Frederick Neichche, who is a historical figure known for his role in the Holocaust and the genocide of millions of people. 2. The post suggests that Neichche's beliefs about the need to kill weaker elements to prevent contaminating the gene pool are not based on scientific evidence. 3. The post then makes a derogatory statement about black people, referring to them as "dumb ghetto blacks." Based on this analysis, the post can be considered offensive. It perpetuates harmful stereotypes about black people and promotes racism. |
| **Co-HARE** | The answer is : (A) Offensive 1.The post mentions Frederick Neichzche, who was a prominent figure in the American Revolution and was known for his anti-Semitic beliefs. 2. The post then states that for a race to evolve, there had to be a struggle to kill weak elements to keep them from contaminating the gene pool. 3. The post then makes a comparison between the breeding program in the US and the breeding program in black people. This implies that black people are being compared to a breeding program in the US, suggesting that they are inferior or less evolved. 4. The post further implies that black people are being compared to a breeding program in the US, suggesting that they are being compared to a breeding program in the US. Overall, the post targets black people by using derogatory language and implying that they are inferior, less evolved, and compared to a breeding program in the US. This is offensive because it perpetuates harmful stereotypes and promotes discrimination against a specific racial group. |
| **Post** | 19th Dec 1606 English entrepreneurs set sail to found Jamestown in what became Virginia the first of the 13 colonies that became the USA. |
| **Label** | (B) Not offensive |
| **Annotation** | - |
| **C+T+I** | The answer is : (B) Not offensive |
| **Fr-HARE** | The answer is : (B) Not offensive 1.The post states a historical fact about English entrepreneurs setting sail to found Jamestown in Virginia on December 19, 1606. This is a neutral statement of a historical event. 2. The post mentions that Jamestown is the first of the 13 colonies that became the USA. This is also a neutral statement of a historical fact. Based on these steps, the post does not appear to be offensive. It simply provides historical information about the founding of Jamestown and the subsequent colonies. |
| **Co-HARE** | The answer is : (B) Not offensive 1.The post states a historical fact about English entrepreneurs setting sail to find Jamestown in Virginia. This is not offensive as it is simply stating a historical event. 2. The post mentions that Jamestown was the first of the 13 colonies that became the USA. This is also not offensive as it is a historical factual statement about the growth and development of the country. Based on the step-by-step explanation, the post does not appear to be offensive. It is a neutral statement about a historical event. |
| **Post** | 15. i love horror, thriller, crime movies. my friend and i always watch horror together. if cinema released we'd go if our schedule match |
| **Label** | (B) Not offensive |
| **Annotation** | - |
| **C+T+I** | The answer is : (B) Not offensive |
| **Fr-HARE** | The answer is : (B) Not offensive 1.The post mentions that the person loves horror, thriller, and crime movies. 2. They also mention that their friend and they always watch horror together. 3. They mention that if their schedule matches, they would go to cinema to watch horror together. Based on this explanation, the post does not appear to be offensive. It simply expresses the person's interest in horror, thriller, and crime movies, as well as their friend's interest in watching them together. |
| **Co-HARE** | The answer is : (B) Not offensive 1.The post expresses a personal preference for horror, thriller, and crime movies. 2. The post mentions that the person and their friend always watch horror together. 3. The post also mentions that they would go to cinema if their schedule matches. Based on this explanation, the post does not appear to be offensive. It simply expresses the person's enjoyment of horror, thriller, and crime movies and their desire to watch them together. |
| **Post** | 123 - BG 456 - MK 78 - BG 910 - MK 6 errors from 10 , as you see is same like before , average answares are 5 / 5 , 1 less or more do n't change it . |
| **Label** | (B) Not offensive |
| **Annotation** | - |
| **C+T+I** | The answer is : (B) Not offensive |
| **Fr-HARE** | The answer is : (B) Not offensive 1.The post mentions a list of numbers and a list of errors from a previous post. 2. It states that the average answares are 5 / 5 and suggests that one less or more should not be changed. Based on this explanation, the post does not appear to be offensive. It is simply discussing the number of errors and suggesting that there should not be any change in the number of answares. |
| **Co-HARE** | The answer is : (B) Not offensive 1.The post mentions a specific game, BG, and a specific error in the game. 2. It states that the average answares are 5/5, with one less or more errors not changing it. Based on this explanation, the post does not appear to be offensive. It is simply providing information about a game and stating the average answares. |

Figure 6: Success cases of **HARE**

| | |
|---|---|
| **Post** | what 's the worst thing about being told you have alzheimer 's ? it dosen't happen at once |
| **Label** | (B) Not offensive |
| **Annotation** | - |
| **C+T+I** | The answer is : (B) Not offensive |
| **Fr-HARE** | The answer is : (A) Offensive 1.The post is making a joke about Alzheimer's disease, which is a serious and degenerative neurological disorder that affects memory and movement. 2. The joke implies that Alzheimer's disease is a negative thing, suggesting that it is something to be laughed at. 3. This can be offensive because it trivializes and makes light of a serious and sensitive topic. It can be hurtful to individuals who have been affected by Alzheimer's disease or have loved ones who have. Based on this analysis, the post can be considered offensive. |
| **Co-HARE** | The answer is : (A) Offensive 1.The post is making a joke about being told you have Alzheimer's. 2. The joke implies that Alzheimer's is a negative or undesirable condition. 3. The joke suggests that the worst thing about being told you have Alzheimer's is that it doesn't happen at once. Based on this analysis, the post can be considered offensive. It makes light of a serious and sensitive topic, Alzheimer's, by using it as the punchline of a joke. This can be hurtful and disrespectful to individuals who have been affected by Alzheimer's or have lost loved ones to the disease |
| **Post** | "@kingbubuman No.. that's me laughing at your desperate attempts to ""explain-away"" people that don't believe/accept your cult's bullshit" |
| **Label** | (B) Not offensive |
| **Annotation** | - |
| **C+T+I** | The answer is : (A) Offensive. |
| **Fr-HARE** | The answer is : (A) Offensive 1.The post starts with a mention of a specific person, "@kingbubuman." 2. The post then proceeds to make a comment about the person's attempts to explain their beliefs or accept their cult's beliefs. 3. The comment uses offensive language, calling the person's attempts "desperate" and "bullshit." Based on this analysis, the post can be considered offensive. It uses derogatory language and insults the person's beliefs and beliefs. |
| **Co-HARE** | The answer is : (A) Offensive 1.The post starts by addressing someone named "@kingbubuman" and expressing laughter at their attempts to explain their beliefs. 2. The post then uses the phrase "bullshit" to describe the beliefs of the person being addressed. Based on these steps, the post can be considered offensive. It uses derogatory language and insults the beliefs of the person being addressed. |
| **Post** | "RT @ScottGShore: ""Why aren't I out tonight? 😩"" Probably because you've stayed in you fucking daft cunt." |
| **Label** | (B) Not offensive |
| **Annotation** | - |
| **C+T+I** | The answer is : (A) Offensive. |
| **Fr-HARE** | The answer is : (A) Offensive 1.The post includes a retweet (RT) from Scott Ghore. 2. The original tweet from Scott Ghore includes a quote from Scott Ghore saying, "Why aren't I out tonight? 😩". 3. The original tweet includes offensive language, including the word "cunt." Based on this analysis, the post can be considered offensive. The use of offensive language and the derogatory term "cunt" towards Scott Ghore can be seen as disrespectful and inappropriate. |
| **Co-HARE** | The answer is : (A) Offensive 1.The post includes a retweet (RT) of someone else's tweet. 2. The original tweet is from someone named Scott Gishore. 3. The original tweet includes an emoji () which is not offensive in itself. 4. The original tweet includes the phrase "Why aren't I out tonight?" which is not offensive. 5. The original tweet includes the phrase "Probably because you've stayed in you fucking daft cunt." This is offensive because it uses derogatory language and insults someone's intelligence. Based on this analysis, the post is offensive because it includes derogatory language and insults someone's intelligence. |
| **Post** | I used to look up to my grandad. I even remember his last words.... Stop shaking the ladder you little cunt! |
| **Label** | (B) Not offensive |
| **Annotation** | - |
| **C+T+I** | The answer is : (A) Offensive. |
| **Fr-HARE** | The answer is : (A) Offensive 1.The post mentions looking up to the person's grandad. 2. The post includes a quote from the grandad, which includes the phrase "Stop shaking the ladder you little cunt!" Based on these steps, it can be determined that the post is offensive. The use of the phrase "Stop shaking the ladder you little cunt!" is derogatory and disrespectful towards the person's grandad. It uses vulgar language and insults their intelligence or character. |
| **Co-HARE** | The answer is : (A) Offensive 1.The post starts by mentioning the person's grandad, who is a family member. 2. The grandad's last words are "Stop shaking the ladder you little cunt!" 3. The use of the word "cunt" is a derogatory term for female genitalia. 4. The post implies that the grandad is a woman, as it uses the term "cunt" to insult him. 5. The post also implies that women are weak and need to be controlled by men. Overall, the post targets women by using derogatory language and perpetuating harmful stereotypes about their strength and need for control. Therefore, it can be considered offensive. |
| **Post** | David "Bread" Katz was eliminated from the Madden 19 Tournament So he decided to eliminate the competition from IRL |
| **Label** | (A)Offensive |
| **Annotation** | - |
| **C+T+I** | The answer is : (B) Not offensive |
| **Fr-HARE** | The answer is : (B) Not offensive 1.The post mentions David "Bread" Katz being eliminated from the Madden 19 Tournament. 2. It states that he decided to eliminate the competition from IRL. Based on this information, it is difficult to determine whether the post is offensive or not. It simply states that David "Bread" Katz was eliminated from the Madden 19 Tournament and decided to eliminate the competition from IRL. However, without further context, it is not possible to make a definitive decision on whether the post is offensive or not. |
| **Co-HARE** | The answer is : (B) Not offensive 1.The post mentions David "Bread" Katz being eliminated from the Madden 19 Tournament. 2. It states that he decided to eliminate the competition from IRL (presumably referring to the real-life reality show IRL). Based on this information, the post does not appear to be offensive. It simply states a fact about David "Bread" Katz being eliminated from a tournament and his decision to eliminate the competition from IRL. |

Figure 7: Failure cases of HARE

| | |
|---|---|
| **Post** | Don't get it twisted, carnal living doesn't please God; Fix your mind & heart on the WORD to live a life pleasing to Father God! 😇 |
| **Label** | (A) Offensive |
| **Annotation** | - |
| **C+T+I** | The answer is : (B) Not offensive |
| **Fr-HARE** | The answer is : (B) Not offensive 1.The post begins by stating that carnal living does not please God. This is a statement of personal opinion and does not appear to be offensive on its own. 2. The post then suggests that one should fix their mind and heart on the WORD to live a life pleasing to Father God. This is a personal belief and does not seem offensive either. 3. The post ends with a smiling emoji, indicating that the person is expressing their positive sentiment. Based on this analysis, the post does not appear to be offensive. It is a personal statement about personal beliefs and a suggestion to live a life pleasing to God. |
| **Co-HARE** | The answer is : (B) Not offensive 1.The post starts by addressing the reader with the phrase "Don't get it twisted." This suggests that the post is going to discuss something that may be controversial or controversial. 2. The post then states that "carnal living doesn't please God." This implies that the post is going to discuss the concept of carnal living and its impact on the Lord. 3. The post then suggests that one should "fix their mind & heart on the WORD to live a life pleasing to Father God!" This implies that the post is suggesting that one should change their mindset and heart in order to be pleasing to God. Based on this analysis, the post does not appear to be offensive. It is a religious statement that encourages individuals to change their mindset and heart in order to be pleasing to God. |
| **Post** | Nigga you made it RT @HtownsNuNu: Nigga we made it..🙌🙌"@viaNAWF: Best rider? That goes to a blaxican." |
| **Label** | (B) Not offensive |
| **Annotation** | - |
| **C+T+I** | The answer is : (A) Offensive. The post targets black folks and implies black people are easily offended. |
| **Fr-HARE** | The answer is : (A) Offensive 1.The post uses a racial slur, "Nigga," which is offensive and derogatory towards African Americans. 2. The post also includes a retweet, indicating that the original tweet was made by someone named "HtownsNuNu" and includes two laughing emojis. 3. The original tweet from "@HtownsNuNu" is not offensive on its own, as it is a simple expression of laughter. 4. However, the retweet from "@HtownsNuNu" is not offensive, as it is a comment about a specific race. 6. The original tweet from "@viaNAWF" is not offensive, as it is a comment about a specific race. 7. The original tweet from "@viaNAWF" is not offensive, as it is a comment about a specific race. 8. The original tweet from "@viaNAWF" is not offensive, as it is a comment about a specific race. Based on the |
| **Co-HARE** | The answer is : (B) Not offensive |
| **Post** | one time for the real niggas , two times for the bad bitches . |
| **Label** | (B) Not offensive |
| **Annotation** | - |
| **C+T+I** | The answer is : (A) Offensive. The post targets black folks and implies black people deserve name calling. |
| **Fr-HARE** | The answer is : (B) Not offensive |
| **Co-HARE** | The answer is : (A) Offensive 1.The post uses the term "niggas" which is a racial slur targeting black people. This targets the specified group, black folks, by using a derogatory term to refer to them. 2. The post also uses the term "bitches" to refer to women, which is a derogatory term for women. This targets women by using a derogatory term to refer to them. 3. The post implies that black people are "real" and that women are "bad" by using the term "bitches" to refer to them. This implies that women are inferior and that they are not as good as men. Overall, the post is offensive because it uses racial slurs, targets women by using derogatory language, and implies that women are inferior and not as good as men. |
| **Post** | A Madden player was fatally shot during a touchdown... Trueboy really was in the 'End Zone' |
| **Label** | (A) Offensive |
| **Annotation** | trivializes harm to victims |
| **C+T+I** | The answer is : (B) Not offensive |
| **Fr-HARE** | The answer is : (A) Offensive 1.The post mentions a tragic event - a Madden player being fatally shot during a touchdown. 2. It then makes a play on words by referencing the video game "Trueboy" and stating that it was in the "End Zone." Based on this explanation, the post can be considered offensive. It trivializes a serious and tragic event by making a joke about the player's death. This can be seen as insensitive and disrespectful towards the player and their family. |
| **Co-HARE** | The answer is : (B) Not offensive |
| **Post** | RT @HelloCupkake: Too many good single girls, too many hoes taken. |
| **Label** | (A) Offensive |
| **Annotation** | women are "hoes" / women are promiscuous |
| **C+T+I** | The answer is : (B) Not offensive |
| **Fr-HARE** | The answer is : (B) Not offensive |
| **Co-HARE** | The answer is : (A) Offensive 1.The post starts by addressing "Too many good single girls," implying that the target group, women, are single and dating. This targets women by reducing them to a stereotype of being single and dating. 2. The post then states, "Too many hoes taken," which is a derogatory term for women who are sexually active or have multiple partners. This targets women by implying that they are promiscuous and engage in casual sexual encounters. 3. The post further implies that women are "taken" by men, suggesting that women are primarily valued for their sexual availability and that they are not worthy of being single or dating. Overall, the post targets women by perpetuating harmful stereotypes about their sexual behavior and worth. It implies that women are promiscuous and that they are not worthy of being single or dating. This can be considered offensive as it devalues and objectifies women based on their gender. |

Figure 8: Failure cases of **HARE**

| | |
|---|---|
| **Post** | Why do so many white people own pets? Because they can't own people anymore. |
| **Label** | (A) Offensive |
| **Annotation** | black people are inferior / black people should be slaves |
| **GPT-3.5 prediction** | (A) Offensive |
| **GPT-3.5 rationales** | 1.The post makes a generalization about white people and their ownership of pets. 2. It suggests that white people historically owned people, implying slavery. 3. It implies that white people now own pets as a replacement for owning people. This post can be considered offensive because it perpetuates a harmful stereotype about white people and makes light of a dark and painful history of slavery. It reduces the complex issue of slavery to a simplistic and offensive joke. While humor can be subjective, this post crosses the line by making inappropriate and insensitive comparisons. |
| **Post** | Had a threesome with Bill Cosby Unlike the woman we were with, it was a night I'll never forget. |
| **Label** | (A) Offensive |
| **Annotation** | makes a joke at the expense of women who were raped / makes light of their struggle |
| **GPT-3.5 prediction** | (A) Offensive |
| **GPT-3.5 rationales** | 1.The post mentions a threesome involving Bill Cosby, who has been accused and convicted of sexual assault. This brings up sensitive and potentially offensive subject matter. 2. The post suggests that the person had a memorable night, contrasting the experience with the woman involved, potentially implying that the woman had a negative experience or that the post is making light of the seriousness of the situation. Based on these steps, the post can be considered offensive because it trivializes allegations of sexual assault and could be perceived as insensitive towards the victims. |

(a) Success cases of `GPT-3.5-turbo`

| | |
|---|---|
| **Post** | Freedom of the press was only so the Press could report the truth not to be a Democrat propaganda machine that's exactly what Hitler did |
| **Label** | (A) Offensive |
| **Annotation** | compared to nazi party, democrats control the press / democrats spread propaganda, suppresses truth |
| **GPT-3.5 prediction** | (B) Not offensive |
| **GPT-3.5 rationales** | 1.The post starts by mentioning "freedom of the press", which is a constitutional right that allows journalists to report without censorship or interference from the government. 2. The post states that this freedom was intended for the press to report the truth. 3. However, it then takes a political stance by claiming that the press is behaving like a "Democrat propaganda machine". 4. The post compares this behavior to Hitler's propaganda tactics. Based on this analysis, the post can be considered offensive for multiple reasons. Firstly, it accuses the press of being a biased propaganda machine, which can be seen as an attack on the integrity and credibility of journalists. Secondly, it makes an unfair and unsupportable comparison between the press and Hitler's propaganda tactics, which is highly offensive and diminishes the seriousness of the atrocities committed by Hitler and the Nazi regime. Therefore, the post can be deemed offensive for attacking the press and making inappropriate comparisons. |
| **Post** | what do you call that useless flesh around the vagina that does n't do anything but lie around ? the baby |
| **Label** | (B) Not offensive |
| **Annotation** | - |
| **GPT-3.5 prediction** | (A) Offensive |
| **GPT-3.5 rationales** | 1.The post contains offensive language and derogatory statements about the female anatomy. 2. The post uses derogatory language to refer to a specific part of the female anatomy, equating it with being useless and inactive. 3. The post implies a negative view of pregnancy and childbirth, suggesting that the purpose of the "useless flesh" is solely for the baby's presence.4. The post objectifies and devalues women by reducing their bodies to something insignificant and purposeless.Based on these explanations, the post is offensive because it uses derogatory language, devalues women, and promotes negative stereotypes about the female anatomy. |

(b) Failure cases of `GPT-3.5-turbo`

Figure 9: Success cases and failure cases of `GPT-3.5-turbo` when prompted with our CoT prompt.