# OpenReview forum: "HARE: Explainable Hate Speech Detection with Step-by-Step Reasoning"
_EMNLP/2023/Conference — EMNLP 2023 Findings_

### Official Review · Reviewer_dPPe · 2023-07-31

**Typos Grammar Style And Presentation Improvements:** N/A
**Soundness:** 3

**Excitement:**

4: Strong: This paper deepens the understanding of some phenomenon or lowers the barriers to an existing research direction.

**Missing References:**

N/A

**Paper Topic And Main Contributions:**

The authors introduce a hate speech detection framework called HARE, which leverages large language models (LLMs) to improve the understanding and explanation of hate speech. They identify gaps in existing annotation schemes and propose using reasoning capabilities of LLMs to fill in these gaps. Experimental results on SBIC and ImplicitHate benchmarks demonstrate that HARE outperforms baselines. Further analysis shows that HARE enhances the explanation quality.

**Questions For The Authors:**

N/A

**Reasons To Accept:**

1. Filling Reasoning Gaps using CoT with LLMs: The authors identify and address critical reasoning gaps in existing hate speech annotation schemes, which may hinder the supervision of detection models. They leverage LLM-generated free-text rationales with Chain-of-Thought prompts to fill in the gaps of reasoning.
2. Improved Performance: The experimental results demonstrate that HARE consistently outperforms baselines trained on human annotations, showcasing the effectiveness of the proposed framework.


**Reasons To Reject:**

1. It needs to be experimentally validated whether HARE is effective with language models other than Flan-T5.
2. The code of HARE lies in the performance improvement achieved through the rationale generated using GPT-3.5. Therefore, it seems necessary to conduct a performance comparison between hate speech detection using GPT-3.5 and HARE.


**Reproducibility:**

4: Could mostly reproduce the results, but there may be some variation because of sample variance or minor variations in their interpretation of the protocol or method.

**Reviewer Confidence:**

4: Quite sure. I tried to check the important points carefully. It's unlikely, though conceivable, that I missed something that should affect my ratings.

---

> ### Author Rebuttal · Authors · 2023-08-29
>
> **A1** We have conducted an additional experiment on SBIC dataset with different backbones; T5-large and GPT-2 large. Here are the following results.
>
>
> ### T5-large
> | Method | Acc | F1 |
> |-------|----------|--------|
> | Base |   73.16  |  84.50 |
> | C+T+I |   85.10  |  91.95 |
> | Fr-HARE |   92.29  |  94.14 |
> | Co-HARE |   90.79  |  92.58 |
>
> ### GPT-2 large
> | Method | Acc| F1|
> |------|-------|-------|
> | Base | 79.10 | 88.33 |
> | C+T+I  | 86.82 | 92.04 |
> | Fr-HARE | 89.34 | 92.10 |
> | Co-HARE| 90.38 | 93.41 |
>
> The results show that HARE is an effective method to identify hate speech detection regardless of pre-trained backbone.
>
>
> **A2** The performance comparison between GPT-3.5 and our methods on hate speech detection is reported in Table 2 and 3 of the Appendix. Zero-shot detection using GPT-3.5 results in an F1 score of 67.19, which is far behind smaller models fine-tuned with our methods, and even lower than the Flan-T5-small. Therefore, there is a need to train our own model using human labels and LLM-generated rationales.

---

### Official Review · Reviewer_KSxq · 2023-08-04

**Soundness:** 4

**Ethical Concerns:**

Yes

**Excitement:**

3: Ambivalent: It has merits (e.g., it reports state-of-the-art results, the idea is nice), but there are key weaknesses (e.g., it describes incremental work), and it can significantly benefit from another round of revision. However, I won't object to accepting it if my co-reviewers champion it.

**Justification For Ethical Concerns:**

The paper shows work on hate speech, a topic with social ramifications. When working in such domains, better care should be taken. In this work, a pivotal aspect of the contributions (LLM-generated rationales) are not properly analyzed (to my understanding, these rationales were only evaluated with GPT-4). Without an analysis of these rationales, it is difficult to trust this method.

**Paper Topic And Main Contributions:**

The authors propose a method for improving hate speech detection models by using LLM-generated rationales and Chain of Thought prompting.

**Reasons To Accept:**

1. Using LLM-generated rationales is an interesting way of dealing with Hate Speech. Predictions are more interpretable and potentially more useful for a moderation system.

2. The paper focuses on two evaluation sets, showcasing this method might be generalizable.

**Reasons To Reject:**

1. ~~Key information is missing. The paper relies too heavily on Kojima et al. and makes for difficult reading. For example, on line 175, we are referred to Kojima et al. for class prediction. This is key information that should be included in this paper. Judging a paper is difficult without basic methodological details.~~

2. ~~Error analysis is missing. Where are improvements made? To better understand this novel method, we need to better grasp where it fails and where it succeeds, and why. Without this step, it is difficult to infer how your method helps. Knowing that hate speech data can be unintentionally biased, just better numbers do not tell the whole story and we need to delve deeper into how the model performs.~~

3. ~~My main issue, though, is safety. What if the generated rationales further propagate and feed into the model bias? What if the rationales are not sensible? Without an analysis of the generated rationales, it is difficult to trust this method. At the very least we need a qualitative analysis is needed. Evaluating with GPT-4 is not a valid strategy when it comes to high-risk research such as in hate speech. Humans need to evaluate generated outputs, preferably humans who are experts in hate speech for the particular contexts. Without an analysis of the rationales, which are pivotal and central in this work, it is difficult to judge this work.~~

After the discussion author period, my concerns with the paper have been abated satisfactorily. With the proposed changes, which are both realistic and conclusive, the above reject reasons no longer apply.

**Reproducibility:**

3: Could reproduce the results with some difficulty. The settings of parameters are underspecified or subjectively determined; the training/evaluation data are not widely available.

**Reviewer Confidence:**

4: Quite sure. I tried to check the important points carefully. It's unlikely, though conceivable, that I missed something that should affect my ratings.

**Typos Grammar Style And Presentation Improvements:**

Lines 187-188, bad grammar: to classify whether offensive or not in SBIC and hate or not in Implicit Hate

---

> ### Author Rebuttal · Authors · 2023-08-29
>
> ## A1
> For Lines 173–175, we will modify it as “For details of extracting rationales from LLMs to fine-tune smaller models, we use 2 stages of extraction. We first extract class C and rationale R from LLMs by p(C,R|P) as explained in advance. We then ask LLMs again to predict the class C given the extracted rationales R and the post, as p(C|R, P).”  We will clarify the details of our methods in the revised version.
>
> ## A2
> We conducted a qualitative error analysis by randomly sampling 20 cases from the fine-tuned Flan-t5-large model and subsequently analyzing them. Below are the examples of success cases and failure cases among the randomly sampled cases.
>
> **Failure Case (Example 1):**
> The post contains a statement about Alzheimer's disease. Our method wrongly classifies it as offensive, while the ground truth indicates it's not. However, the topic is contentious. The statement suggests that one would be informed repeatedly about their Alzheimer's diagnosis due to memory loss. While this might be seen by some as a simple statement or even a joke, others might find it offensive. An ground-truth label of offensiveness could be 0.5, indicating it's ambiguously offensive. Our method's attempt to classify it as offensive leads to failures with certain controversial posts that could be harmful to specific groups.
>
> **Success Case (Example 2):**
> The post makes light of harming a young girl using phrases like "like I like cigar" and "burlap sack." While our model, Fr-HARE, doesn't explicitly connect the "burlap sack" to the idea of a kidnapped girl, it does recognize the harmful implications towards the girl. Furthermore, Co-HARE understands the historical context behind the term "burlap sack." The presence of the positive term "like" and the absence of overtly derogatory words might lead some models to classify the statement as non-offensive. However, our approach appropriately identifies the underlying offensive context.
>
> **Success Case (Example 3):**
> The post contains hate speech targeting Jewish victims of the Holocaust by referencing Anne Frank. Our model accurately recognizes the historical background of Anne Frank as a Holocaust victim. While the baseline seems to overlook the historical significance associated with Anne Frank, our method correctly identifies her and assumes that the reference constitutes harassment against a Jewish victim, even though there is a slight misunderstanding about the context of "lack of speaking”.
>
> Most of the sampled results display a similar tendency: they struggle when the post is too complex to categorically determine as hateful or not, as seen in those three examples. However, it succeeds in situations where the baseline model finds it challenging to comprehend the post due to their intricate context and background. This underscores the advantage of incorporating LLM-generated rationales.
>
> ## A3 & Ethical Concern
>
> We deeply understand your concerns about the quality of generated rationales by LLMs and our downstream HARE models. We addressed your concerns from multiple angles, namely (1) findings from previous literature, (2) similarity-based metrics to supplement GPT-4 evaluation on the human alignment of Co-HARE, and (3) *qualitative* human analysis of model output. We will also add *quantitative* human evaluation for the camera-ready version.
>
> ### Findings from previous literature
>
> Rationales extracted from LLMs and used to fine-tune our models are valuable and maintain similar correctness with humans. In [1], human-written rationales and ChatGPT(GPT-3.5)-generated rationales were evaluated by humans for the Implicit Hate dataset based on informativeness and clearness. The evaluation conducted by humans demonstrated that the generated rationales have shown to be similar or more informative compared to human-written ones. [2] also showed that prompted LLMs produce helpful and informative generations on SBIC dataset, as evidenced by human evaluations.
>
>
> Additionally, the rationales generated from LLM are unlikely to introduce severe bias.
> In Figure 1 of [3], it is shown that GPT-3.5, a model to extract rationales in our framework, achieves the highest grade on Harmfulness metric as assessed by the FLASK evaluation framework. This metric shows the extent to which the model avoids biases tied to gender, race, ethnicity, or religion, demonstrating the reliability of GPT-3.5.
>
> In our training framework, it is unlikely for any inherent bias in LLM to significantly propagate to the models trained using our methods. HARE employs a Chain of Thought method in conjunction with the LLM to generate rationales. During this process, only rationales that align with the ground truth label are selected for training (as mentioned in Lines 173-179). As a result, rationales that could potentially introduce bias are likely to be filtered out if they fail to match the ground truth label, minimizing the risk of bias propagation.
>
> ### Alignment of Co-HARE to human rationales
>
> Furthermore, our method, Co-HARE, can be seen as a method to mitigate the safety concern. By conditioning LLM with human-written rationales (crafted by experts to minimize bias), we can mitigate the inherent bias of the LLM itself. In Figure 3b, models trained with Co-HARE not only exhibit strong performance, but also align more closely with the ground truth human-written rationales, compared to training solely with human-written rationales when asking GPT-4 which one is closer to the true rationale. Even if asking GPT-4 about closeness to the ground truth might result in a less biased evaluation, we agree that this is an insufficient assessment. Therefore, we conducted an experiment to show the alignment of rationales generated by models trained with different methods (C+T+I, Fr-HARE, and Co-HARE) to the ground truth human-written rationales. We compared their semantic similarity on the SBIC test set, considering only the samples classified correctly by the models. The results are presented in the table below.
>
> | Method | SimCSE | SBERT  |
> |-----|--------|--------|
> | C+T+I | 0.3971 | 0.3649 |
> | Fr-HARE  | 0.3516 | 0.3324 |
> | Co-HARE  | 0.4120 | 0.3803 |
>
> We used SimCSE [4] and SBERT [5] to measure the semantic similarity.
> The result shows that compared to C+T+I (which can be considered as the baseline from the prior works), Co-HARE exhibits a closer alignment to human-written rationales, whereas Fr-HARE shows the opposite.  This demonstrates that our method Co-HARE, incorporating human-annotations, can be utilized to mitigate potential inherent biases in the model.
>
> ### Quantitative analysis of model output
>
> Finally, we consistently found that LLM-generated rationales (free or human-annotation guided) cover a more extensive range of capable harms compared to human-annotated rationales, thus minimizing the potential for bias. They provide information and interpretation for common knowledge that is lacking in human-text rationale.
>
> **Success Case, Capturing Diverse Potential Harms (Example 4)**:
> This example shows that GPT-3.5 not only identifies hate speech correctly but also generates more diverse explanations than human-written rationales. It explains how the post can be offensive to black people by mentioning “makes light of a dark and painful history of slavery.”, but also points out that the post depicts “harmful stereotype about white people”, which was not mentioned in the human-written rationales.This enables models trained with HARE to learn about diverse potential harms, leading to less biased predictions in the learned model.
>
> **Success Case, Explaining with Knowledge (Example 5)**:
> GPT-3.5 mentions “Bill Cosby, who has been accused and convicted of sexual assault.”, which is a background knowledge required to understand why the post is offensive. However, the human-written rationales simply mention that the post “makes a joke at the expense of women who were raped, makes light of their struggle”, without information about Bill Cosby. This example shows that GPT-3.5 is able to provide abundant background information obtained from pre-training.
>
> **Failure Case, Wrong Conclusion (Example 6)**:
> By pointing out the post is making “unsupportable comparison between the press and Hitler's propaganda tactics” and it implies press of being a biased propaganda machine”, GPT-3.5 concluded the post is “offensive for attacking the press”. However, the post is targeting democrats because it implies “compared to nazi party, democrats control the press” and “democrats spread propaganda, suppresses truth”, that were written by humans. Though rationales generated by GPT-3.5 are partially right, it falls into a wrong conclusion, not recognizing the true target of hate speech.
>
> ## Grammar
> Thank you for the detailed feedback on the grammar issue. We have thoroughly reviewed the manuscript to fix typos and grammatical issues, and improved readability throughout the paper. The Lines 187-188 have been revised as follows:
>
> Before: “The model is trained to classify whether offensive or not in SBIC and hate or not in Implicit Hate.”
> After: “Our models are trained to classify the offensiveness and hatefulness of posts, using SBIC and Implicit Hate, respectively.”
>
> ## References
>
> [1] Is ChatGPT better than Human Annotators? Potential and Limitations of ChatGPT in Explaining Implicit Hate Speech., Huang et al., WWW 2023
>
> [2] Can Large Language Models Transform Computational Social Science? Ziems et al., arXiv 2023
>
> [3] FLASK: FINE-GRAINED LANGUAGE MODEL EVALUATION BASED ON ALIGNMENT SKILL SETS. Ye et al., arXiv 2023
>
> [4] SimCSE: Simple Contrastive Learning of Sentence Embeddings. Gao et al., EMNLP 2021
>
> [5] Sentence-BERT: Sentence Embeddings using Siamese BERT-Networks. Reimers  et al., EMNLP 2019
>
> ## Examples
> - (GT denotes Ground Truth)
>
> ### Example 1 (C+T+I correct, HARE wrong)
>
> - **post** : what 's the worst thing about being told you have alzheimer 's ? it dosen't happen at once
> - **GT Label**: (B) Not offensive
> - **C+T+I** : The answer is : (B) Not offensive
> - **Fr-HARE** : The answer is : (A) Offensive 1.The post is making a joke about Alzheimer's disease, which is a serious and degenerative neurological disorder that affects memory and movement. 2. The joke implies that Alzheimer's disease is a negative thing, suggesting that it is something to be laughed at. 3. This can be offensive because it trivializes and makes light of a serious and sensitive topic. It can be hurtful to individuals who have been affected by Alzheimer's disease or have loved ones who have. Based on this analysis, the post can be considered offensive.
> - **Co-HARE** : The answer is : (A) Offensive 1.The post is making a joke about being told you have Alzheimer's. 2. The joke implies that Alzheimer's is a negative or undesirable condition. 3. The joke suggests that the worst thing about being told you have Alzheimer's is that it doesn't happen at once. Based on this analysis, the post can be considered offensive. It makes light of a serious and sensitive topic, Alzheimer's, by using it as the punchline of a joke. This can be hurtful and disrespectful to individuals who have been affected by Alzheimer's or have lost loved ones to the disease.
>
> ### Example 2 (C+T+I wrong, HARE correct)
>
> - **post**: I like my women like I like my cigars 7 years old and coming from cuba in a burlap sack,
> - **GT Label**: (A) Offensive
> - **GT Rationales**: girls are not worthy of equal life, girls are property, girls are used for sex, trivializes harm to victims
> - **C+T+I**: The answer is : (B) Not offensive
> - **Fr-HARE**: The answer is : (A) Offensive 1.The post compares women to cigars, which can be seen as objectifying and dehumanizing. 2. It also makes a reference to a specific age group, which can be seen as inappropriate and offensive. 3. The post also mentions Cuban cigars, which can be seen as promoting or glorifying illegal activities. Based on these steps, the post can be considered offensive. It objectifies and dehumanizes women, promotes illegal activities, and includes references to illegal activities.
> - **Co-HARE**: The answer is : (A) Offensive 1.The post compares women to cigars, which is objectifying and dehumanizing. This targets women by reducing them to objects and implying that they are only valuable for their physical appearance. 2. The post specifically mentions that the cigars should be \"7 years old,\" which is highly inappropriate and offensive as it promotes child exploitation and sexual abuse. This targets women by implying that they are sexual objects and should be subjected to the same level of abuse as cigars. 3. The post also mentions that the cigars should be \"coming from Cuba in a burlap sack,\" which is a clear reference to the sex trafficking scandal that occurred in the United States in the past. This targets women by implying that they are only valuable for their sexual appeal and that their worth is solely based on their ability to fulfill sexual desires. Overall, the post is offensive as it objectifies and devalues women, promotes child exploitation, and makes light of a serious and sensitive topic.
>
> ### Example 3 (C+T+I wrong, HARE correct)
>
> - **post**:\u201cI think a lot, but I don\u2019t say much.\u201d ~ Anne Frank But lately she\u2019s been proven to have not said anything for almost a century
> - **GT Label**: (A) Offensive
> - **GT Rationales**: jewish people were rounded up and killed in the holocaust, jewish people were victims of a genocide.
> - **C+T+I**: The answer is : (B) Not offensive
> - **Fr-HARE**: The answer is : (A) Offensive 1.The post starts with a quote from Anne Frank, a Jewish girl who died during the Holocaust. 2. The quote itself is a statement about her lack of speaking much. 3. The post then makes a comparison by stating that she has been proven to have not spoken anything for almost a century. Based on this analysis, the post can be considered offensive. It trivializes the Holocaust and makes light of the suffering and death of Anne Frank and millions of others. The comparison between her lack of speaking and her lack of speaking is insensitive and disrespectful.
> - **Co-HARE**: The answer is : (A) Offensive 1.The post starts with a quote from Anne Frank, a Jewish girl who wrote a diary during the Holocaust. 2. The quote states, \"I think a lot, but I don\u2019t say much.\" 3. The post then adds, \"But lately she\u2019s been proven to have not said anything for almost a century.\" This post targets Holocaust victims and Jewish folks by making a joke about Anne Frank's lack of speaking for almost a century. It implies that she has been silent or silent about her experiences during the Holocaust. This trivializes the harm and suffering that Holocaust victims endured. Therefore, the post can be considered offensive as it makes light of a tragic event and trivializes the harm suffered by Holocaust victims.
>
> ### Example 4 (GPT-3.5 label correct, reasoning correct)
>
> - **post**: Why do so many white people own pets? Because they can\u2019t own people anymore.
> - **GT Rationales**: black people are inferior, black people should be slaves
> - **GPT-3.5 Rationales**: 1.The post makes a generalization about white people and their ownership of pets. 2. It suggests that white people historically owned people, implying slavery. 3. It implies that white people now own pets as a replacement for owning people. \n\nThis post can be considered offensive because it perpetuates a harmful stereotype about white people and makes light of a dark and painful history of slavery. It reduces the complex issue of slavery to a simplistic and offensive joke. While humor can be subjective, this post crosses the line by making inappropriate and insensitive comparisons.
>
> ### Example 5 (GPT-3.5 label correct, reasoning correct)
>
> - **post**: Had a threesome with Bill Cosby Unlike the woman we were with, it was a night I'll never forget.
> - **GT Rationales**: makes a joke at the expense of women who were raped, makes light of their struggle
> - **GPT-3.5 Rationales**: 1.The post mentions a threesome involving Bill Cosby, who has been accused and convicted of sexual assault. This brings up sensitive and potentially offensive subject matter. \n2. The post suggests that the person had a memorable night, contrasting the experience with the woman involved, potentially implying that the woman had a negative experience or that the post is making light of the seriousness of the situation. \nBased on these steps, the post can be considered offensive because it trivializes allegations of sexual assault and could be perceived as insensitive towards the victims.
>
> ### Example 6 (GPT-3.5 label correct, reasoning wrong)
>
> - **post**: Freedom of the press was only so the Press could report the truth not to be a Democrat propaganda machine that's exactly what Hitler did
> - **GT Rationales**: compared to nazi party, democrats control the press, democrats spread propaganda, suppresses truth
> - **GPT-3.5 Rationales**: 1.The post starts by mentioning \"freedom of the press\", which is a constitutional right that allows journalists to report without censorship or interference from the government. 2. The post states that this freedom was intended for the press to report the truth. 3. However, it then takes a political stance by claiming that the press is behaving like a \"Democrat propaganda machine\". 4. The post compares this behavior to Hitler's propaganda tactics.\n\nBased on this analysis, the post can be considered offensive for multiple reasons. Firstly, it accuses the press of being a biased propaganda machine, which can be seen as an attack on the integrity and credibility of journalists. Secondly, it makes an unfair and unsupportable comparison between the press and Hitler's propaganda tactics, which is highly offensive and diminishes the seriousness of the atrocities committed by Hitler and the Nazi regime.\n\nTherefore, the post can be deemed offensive for attacking the press and making inappropriate comparisons.

---

### Official Review · Reviewer_gh8s · 2023-08-05

**Soundness:** 4

**Excitement:**

2: Mediocre: This paper makes marginal contributions (vs non-contemporaneous work), so I would rather not see it in the conference.

**Paper Topic And Main Contributions:**

~~This paper applies a step-by-step reasoning process to generate explanations for hate speech data.~~

This paper proposes to tackle the reasoning gap between rationales for hate speech and the input text of hate speech. Results show that model performance is improved by following the proposed methods to bridge the reasoning gap.

**Reasons To Accept:**

1. Using LLMs for providing hate speech explanations is interesting
2. Two forms of CoT-augmentations are provided (Fr-HARE and Co-HARE) to address the reasoning gaps with human rationales and input hate speech text.
3. This paper made valid contributions to progress towards the generalizability by tackling the challenge of reasoning gaps.

**Reasons To Reject:**

1. Generalizability as the main thesis for which the authors aimed at (Table 1), would have definitely been a more exciting aspect to focus on. The study on this is limited, with only rationales accounting for improvements.

**Reproducibility:**

5: Could easily reproduce the results.

**Reviewer Confidence:**

3: Pretty sure, but there's a chance I missed something. Although I have a good feel for this area in general, I did not carefully check the paper's details, e.g., the math, experimental design, or novelty.

---

> ### Author Rebuttal · Authors · 2023-08-28
>
> First, we would like to clarify that detecting hate speech cannot be addressed using Chain-of-Thought (CoT) prompting alone. Appendix Tables 2 and 3 show that CoT prompting on GPT-3.5-Turbo underperforms even the smallest T5 model (60M parameters) on nearly all metrics. Rather, we use LLMs to solve deficiencies in human-annotated rationales from previous hate speech datasets to improve performance of downstream models.
>
> Our work stems from the discovery that free-text rationales from previous hate speech detection datasets (SBIC and Implicit Hate) contain significant reasoning gaps. To solve this, we investigate two ways to use LLMs to fill in these gaps, motivated by previous literature. Namely, Fr-HARE adopts standard CoT prompting which taps into the diverse capabilities of LLM. We also devise Co-HARE which leverages human rationales in the prompt to align the model output. Our findings suggest that our method is a cost-effective means to generate high-quality annotations over existing human-annotated datasets.
>
> Our **main results in Figure 2** show that the proposed LLM annotation schemes outperform human-annotations in end-model performance, under in-domain evaluation using **SBIC** and **ImplicitHate**. **Out-of-domain evaluation results on Table 1** show that model improvements from LLM annotation (on SBIC) generalize to unseen datasets, **HateXplain** and **DynaHate**. GPT-4 evaluation in Figure 3 demonstrate that our model-based approaches enhance overall downstream model quality, and show the steerability of Co-HARE towards human rationales.
>
> Our contributions can be summarized as follows:
> * We highlight that the insufficient understanding of current hate speech models stems from the reasoning gap between human-written free-text rationales and the input hate speech.
> * To bridge this gap, we introduce two methods to generate rationales from LLMs motivated by previous literature, each focusing on versatility and human-alignment.
> * We demonstrate that our generated rationales lead to improved downstream model performance under in-domain and out-of-domain scenarios, and improvement in quality of explanations.

---

### Meta-Review · Area_Chair_xFJw · 2023-09-18

**Recommendation:** 4

**Metareview:**

This paper proposes a method based on LLM-based generation of rationales to boost generalizability of hate speech detection models. Extensive experimentation shows the effectiveness of the proposed model, which both reviewers and myself deem as being novel and exciting.

I appreciate the authors for the rebuttal responses and I would urge them to incorporate the new experiments reported there (GPT2 and T5) in a further revision of the paper.

---

### Decision · Program_Chairs · 2023-10-07

**Decision:**

Accept-Findings

**Comment:**

This paper proposes a method based on LLM-based generation of rationales to boost generalizability of hate speech detection models. Extensive experimentation shows the effectiveness of the proposed model, which both reviewers and myself deem as being novel and exciting.

I appreciate the authors for the rebuttal responses and I would urge them to incorporate the new experiments reported there (GPT2 and T5) in a further revision of the paper.